# Robust cardiac segmentation corrected with heuristics

**Alan Cervantes-Guzmán**[1☯], **Kyle McPherson**[3☯], **Jimena Olveres**[1,2‡], **Carlos Francisco Moreno-García**[3‡], **Fabián Torres Robles**[1], **Eyad Elyan**[3], **Boris Escalante-Ramírez**[1,2]*

**1** Facultad de Ingenieria, Universidad Nacional Autonoma de Mexico, Mexico City, Mexico, **2** Centro de Estudios en Computacion Avanzada, Universidad Nacional Autonoma de Mexico, Mexico City, Mexico, **3** School of Computing, Robert Gordon University, Aberdeen, United Kingdom

☯ These authors contributed equally to this work.
‡ JO and CFMG also contributed equally to this work.
* boris@unam.mx

**Data Availability Statement:** Reference 39 in the manuscript contains a link to the public data base used in this work, namely: https://aimi.stanford.edu/echonet-dynamic-cardiac-ultrasound. Furthermore, a sample data set of 100 images

## Abstract

Cardiovascular diseases related to the right side of the heart, such as Pulmonary Hypertension, are some of the leading causes of death among the Mexican (and worldwide) population. To avoid invasive techniques such as catheterizing the heart, improving the segmenting performance of medical echocardiographic systems can be an option to early detect diseases related to the right-side of the heart. While current medical imaging systems perform well segmenting automatically the left side of the heart, they typically struggle segmenting the right-side cavities. This paper presents a robust cardiac segmentation algorithm based on the popular U-NET architecture capable of accurately segmenting the four cavities with a reduced training dataset. Moreover, we propose two additional steps to improve the quality of the results in our machine learning model, 1) a segmentation algorithm capable of accurately detecting cone shapes (as it has been trained and refined with multiple data sources) and 2) a post-processing step which refines the shape and contours of the segmentation based on heuristics provided by the clinicians. Our results demonstrate that the proposed techniques achieve segmentation accuracy comparable to state-of-the-art methods in datasets commonly used for this practice, as well as in datasets compiled by our medical team. Furthermore, we tested the validity of the post-processing correction step within the same sequence of images and demonstrated its consistency with manual segmentations performed by clinicians.

## 1. Introduction

Medical images acquired through several modalities are useful to study and analyze anatomic information and improve medical diagnosis. To this end, anatomical structures must be isolated to evaluate them in detail, therefore segmentation is one of the most important tasks in medical imaging which allows obtaining qualitative and quantitative information relevant to clinical specialists [1–3]. However, medical imaging quality is impaired by several factors such as limited spatial resolution, noise and low contrast (to name a few) which makes

**Funding:** Newton Fund Institutional Links program project 527639907 granted by The British Council, UK, https://www.britishcouncil.org/education/he-science/newton-fund (A.C., K. M., J.O., C.F.M., F.T.R., E.E., B.E.) Secretaría de Educación, Ciencia, Tecnología e Innovación, CDMX, under grant 202/2019 (A.C., K. M., J.O., C.F.M., F.T.R., E.E., B.E.) https://www.sectei.cdmx.gob.mx/ Universidad Nacional Autónoma de México PAPIIT grant IV100420 https://dgapa.unam.mx/index.php/impulso-a-la-investigacion/papiit (A.C., J.O., F.T.R., B.E.) Universidad Nacional Autónoma de México Programa de Becas Posdoctorales https://dgapa.unam.mx/index.php/formacion-academica/posdoc (F.T.R., B.E.) The funders had no role in study design, data collection and analysis, decision to publish, or preparation of the manuscript.

**Competing interests:** The authors have declared that no competing interests exist.

segmentation a complicated task. Traditionally, segmentation of anatomic structures requires the assessment of a specialist, which is a tedious task and specialist dependent and, therefore prone to errors and inaccuracies [4, 5].

Most segmentation algorithms for cardiac imaging found in the literature are applied to magnetic resonance images (MRIs) [6]. Due to its good contrast this imaging modality produces very good results for clinicians. Computerized tomography (CT) images do not offer as good contrast as MRI. However, CT is more accessible and has enough resolution to distinguish adjacent organs [7, 8]. Furthermore, echocardiographic (i.e. ultrasound) imaging systems are widely available due to their low cost and portability. Their main disadvantage is the high correlated noise present in the images called speckle. This type of noise leads to the need of a well-trained specialist to discriminate anatomical structures from noise, which can influence diagnosis results. This often turns into bias and dependence on the equipment's operator to acquire the correct image [9].

Despite the aforementioned disadvantages, echocardiography is the most non-invasive method used to analyse cardiac cavities since it delivers real-time images in an accessible and portable way [10]. Most recently, deep learning architectures such as convolutional neural networks (CNN) have been successfully applied in medical image analysis [11]; however, they are often trained to analyse only the left ventricle. New attempts to improve these systems are made daily, focusing on characterizing the segmentation of more than one cardiac cavity. [12] However, despite the existence of works such as [13] devoted to four cardiac chambers segmentation in fetal echocardiography, there are no results in the state-of-art addressing the problem of four cardiac chamber segmentation in adult echocardiography images.

Furthermore, the American Society of Echocardiography and the European Association of Cardiovascular Imaging provide a set of guidelines for assessing measurements related to the four cardiac chambers. They state that these measurements are essential for evaluating cardiac function and extracting important clinical parameters [7].

In this paper, we present a robust cardiac segmentation tool that not only segments the heart cavities in ultrasound images but is also robust to noise and text insertions, common in these studies. Furthermore, this tool also pre and post processes segmentations in order to detect the heart within the cone-shape area of B-scans and refines segmentations by means of specialized heuristics, which matches clinical-expert criteria.

## 2. Related work

Multiple methods for segmentation tasks in medical images have been proposed over the years, deformable models such as Active Contour Models (ACM) [14] and Active Shape Models (ASM) [15] being some of the most popular. They have been widely employed for such tasks, even in recent times. Some of these studies [16, 17] use deformable models to segment cardiac medical images in different medical imaging modalities. However, deformable models possess certain limitations such as the need of a good initialization of the shape to be segmented. Especially in the case of noisy images, like echocardiography images, the contours often do not manage to converge to the desired outline [18].

In recent years, the development of deep learning models that automatically perform a wide range of tasks has increased exponentially. Medical tasks such as image segmentation have also improved because of their use [19, 20]. One of the most important and recent models for biomedical image segmentation is called U-Net, which was initially presented by Ronneberger et al. [21]. The name of this method comes from its end-to-end U-shape, aligned with the architecture of a fully convolutional network (FCN). This network achieves a very precise semantic segmentation, requiring fewer annotated images than other CNN-based

architectures, alongside data augmentation. The U-Net architecture is an encoder-decoder type architecture that consists of two parts: a contracting path where an ordinary convolutional process happens and the expansive path constituted by transposed convolutional layers. Despite being published more than five years ago, this model is very relevant nowadays and has been used in multiple applications [22–25].

One of the most recent and updated surveys on cardiac segmentation presented by Chen et. al [26] shows the predominance and versatility of U-Net as a viable segmentation algorithm in this domain, given that from 77 works reported, 25 of them use U-Net [21]. Meanwhile another survey of recent advances and clinical applications of deep learning in medical image analysis presented recently in 2021 by Chen et. al [27] also shows the same predominance and versatility of U-Net (and its variants) in multiple medical image segmentation tasks with different medical image modalities (20 works used U-Net or one of its variants from 27) maintaining the relevance of this model.

Some of the works presented in the cardiac segmentation survey [26] are entirely focused on cardiac ultrasound segmentation using U-Net or combining it with other models, such as Deformable Models, Kalman filter-based methods and other deep learning architectures such as TL-Net [28]. Chen presents a compilation of the segmentation methods used up to 2020 on the cardiac anatomical structures of medical interest [26].

Although there are several variants of the U-Net, classic U-Net architecture [21] continues being relevant for medical image segmentation tasks. This is demonstrated by the benchmarking research made by Gut et. al. [29] in 2022 where the performance of the classic U-Net was compared against its variants such as UNet++, ResUNet, CPFNet, CS2-Net and UNet 3+ in 9 different medical image segmentation tasks. All the models were evaluated with several metrics resulting in classic U-Net as the model with the less training and inference time with a higher memory efficiency than its variants.

In 2019, Leclerc et al. [30] analyzed the performance of different models to segment different structures of the left ventricle on the apical four-chamber echocardiography plane. The models were a U-Net optimized for speed, a U-Net optimized for accuracy, and U-Net++. Other deep learning models are also compared, such as a Neural Network which uses prior anatomical information to improve image segmentation known as an Anatomically Constrained Neural Network (ACNN) [31] and the Stacked Hourglasses (SHG) encode-decode network, a network based on the successive steps of downsampling layers (using pooling methods) and upsampling layers to produce a final set of predictions [32]. Finally, non-deep learning methods such as Structured Random Forest (SRF), B-Spline Explicit Active Surface Model (BEASM-full mode) and B-Spline Explicit Active Surface Model (BEASM-semi mode) are mentioned by Leclerc. A public dataset called CAMUS (Cardiac Acquisition for Multi-structure Ultrasound Segmentation) was used for this segmentation purpose [33]. The results of the experiments with the deep learning-based methods were better than the non-deep learning ones, being the U-Net optimized for accuracy the best of all.

Rachmatullah et al. also used U-Net methods on standard fetal images [34] from ultrasound data obtained from videos. They also employed post-processing methods to enhance their results. Yin et al. [35] addressed current challenges in medical image segmentation, showing how different authors tackle the problem with different U-Net networks and collecting some experiments using these algorithms. Most recently, Dang et al. [36, 37] implemented the study of a weighted ensemble of deep learning methods based on Comprehensive Learning Particle Swarm Optimization (CLPSO) for cardiac segmentation task. To this end, they trained six transfer learning models for segmentation, which were then assembled to get the best possible output. This output is calculated as the weighted sum of segmentation outputs, and the CLPSO algorithm is used to optimize the combined weights. These transfer learning systems

were retrained using the CAMUS dataset, which contains 250 images of hearts where only the LV and the LA were segmented.

One of the key drawbacks that must be highlighted from the current state-of-the-art ultrasound image analysis is that most of the efforts are focused on left ventricle image segmentation. For instance, another well-known documented database was collated by Ouyang et al. [38]. They segment the heart's left ventricle and predict the ejection fraction calculus to classify heart failure. They claim that their variance is similar to that of human experts. But, notably, they make available their dataset consisting of annotated echocardiogram videos. To our knowledge, very few efforts have been made to segment the four chambers from a four-chamber view echocardiogram video since this task is quite difficult even for an expert human eye.

## 3. Materials and methods

In this study, we used the Database EchoNet-Dynamic, a dataset provided for the Center of Artificial Intelligence in Medicine and Imaging from Stanford University [39]. Furhtermore, clinicians from the medical center "20 de Noviembre" in Mexico City provided 120 sequences with the four chambers segmented so that the systems could be retrained once again and were capable of localizing all four heart chambers.

For the methodology, we implemented a process on the ultrasound images that includes cone segmentation, a four-chamber segmentation including the left and right ventricle and left and right atrium. Although the method shows promising results and higher accuracy rates compared to state-of-the-art results, there were some noticeable errors in some results, such as segmentation leakage in one of the four chambers detected and sometimes of irregular shape. This was the reason to add a final step for error correction.

### 3.1 Cone segmentation

One of the main reasons for poor performance in cardiac segmentation approaches is that, in practice, systems must deal with low-quality and previously annotated images. Fig 1 shows one of these cases obtained from a specialist clinic in Mexico. This image shows a cardiac cycle screen captured from the measurement device by the clinician. Therefore, not only is the quality poor but there are also annotations around the image (i.e. text, patient's data, date of measurement, cardiac cycle signal), which result in artifacts that hinder the analysis. Therefore, the first step of the present method is to create a cone segmentation class which can detect the position and location of the central cone and use this for further stages. Implementing a cone segmentation module not only improves the accuracy of the chamber segmentation but also reduces the human effort of manually cutting out the cone. In addition, we trained this model with masks from different ultrasound images (i.e. fetal, abdominal, among others), and thus, our approach is effective for any ultrasound cone segmentation.

For the segmentation task, we use Detectron2, a popular library developed by Facebook AI Research, to implement a Mask R-CNN [40] with ResNet-50 [41] and a Feature Pyramid Network [42] (FPN) as the backbone. The former is a popular and effective CNN architecture used in computer vision tasks, with 50 layers that use residual connections to overcome the problem of vanishing gradients throughout training. At the same time, the latter helps to leverage the pre-trained weights from ImageNet by improving the representation of the detected objects at different scales. The model is trained with varying epochs, specifically 100, 350, 500, 1000, and 1500 epochs, which are subsequently evaluated to determine the optimal choice, after which five k-fold cross-validation is used. The method developed can deal with any input (e.g. .jpg, .png, .avi, DICOM files, etc.). Furthermore, smoothing and dilation can be used to improve the output of the predicted mask.

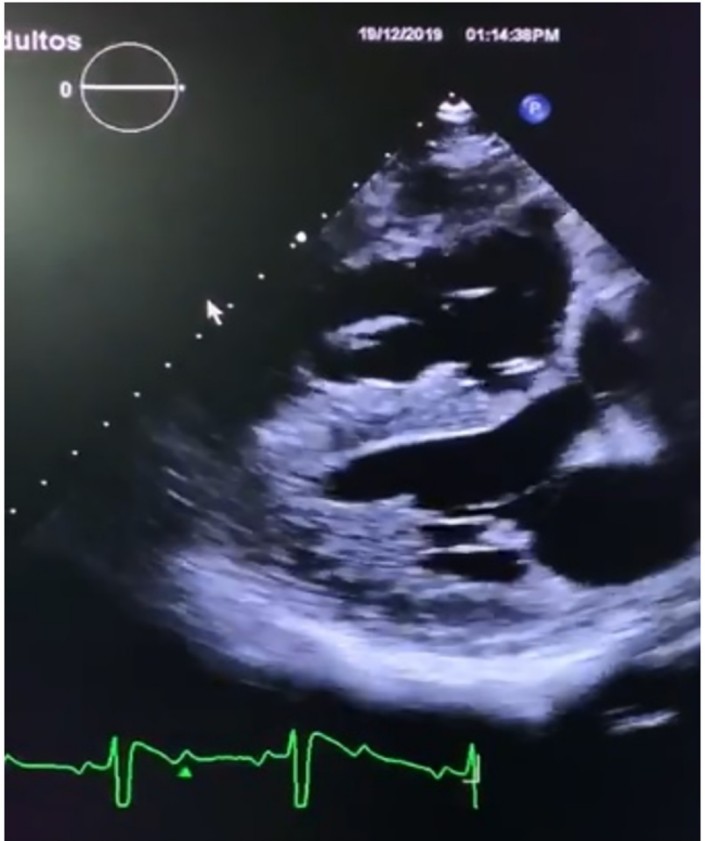

**Fig 1. An example of a low resolution, annotated echocardiographic image.**

Fig 2 shows an example of the cone segmentation task applied on a renal liver ultrasound. The original image can be seen in the upper left corner. The scan segmentation module is used, which segments the cone (upper-right corner). Notice that the jagged edges of the mask produce an irregular shape on the cone. This can be corrected by smoothing the mask to maintain the regular shape of the cone beam as it gets applied to the image in the the bottom-left corner. The final result is shown in the bottom-right corner after applying dilation to the mask to improve the corners of the cone beam.

## 3.2 U-Net

As mentioned earlier in the introduction section, the main limitation of medical image analysis resided in the need for a large and reliable labeled training dataset. In previous years, this requirement seemed to have become stricter, especially as deep learning research has focused on medical image segmentation tasks. Most recently, there has been an increase in the amount of publicly available medical datasets for cardiac segmentation [26] and also, of deep learning algorithms to handle them, which have derived from a higher number of publications related to image segmentation. In fact, methods such as U-Net have been emphasized to work on a reduced data set [21]. For this reason, we have selected U-Net as the basis of the deep learning architecture that will be implemented for our segmentation task.

Fig 3 shows the U-Net architecture implemented, this architecture follows quite the original architecture from Ronnerberger's work, the number of contraction and expansion blocks,

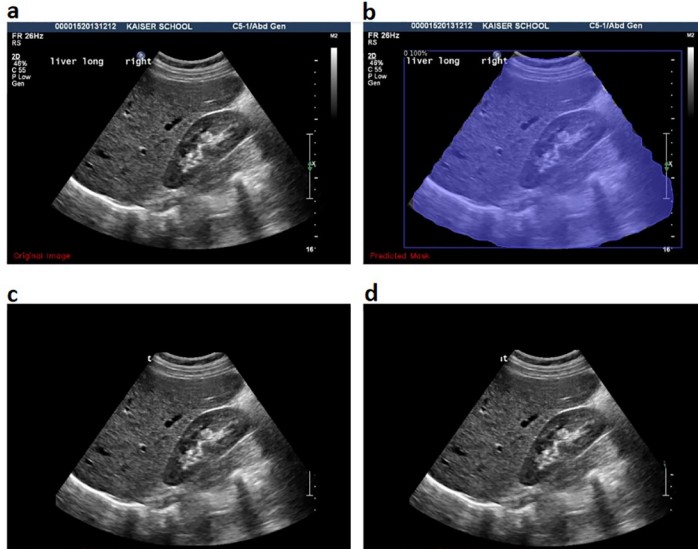

**Fig 2. An example of the cone segmentation task executed in a renal liver ultrasound. a**, the original image features all the technical elements of the ultrasound scan. **b**, the predicted mask from the cone segmentation superimposed on the ultrasound scan, **c**, the mask smoothed and applied over the image. **d**, finally a dilation step is applied to improve the cone segmentation.

even the bottleneck remains, but, as a complement, a batch normalization module was added at the end of each convolutional layer. This batch normalization module helps the model to get an easier initialization of parameters, to get a faster training and even to reduce the very popular trouble of overfitting. Table 1 shows a summary of whole the architecture of our U-Net, and also includes the number of inner parameters. The machine learning model was implemented using Python language along PyTorch, the optimized tensor library for deep learning.

### 3.3 Heuristic correction

Two main reasons led us to implement a heuristics correction module after the cone segmentation task. Firstly, the UNET predicted more than four heart chamber masks for certain frames in the sequence. Secondly, some frames in the sequence showed overlapping heart

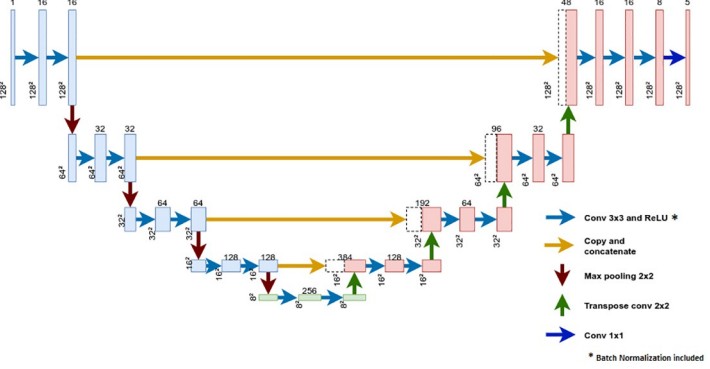

**Fig 3. U-Net architecture implemented.**

**Table 1. Architecture summary of U-Net implemented, parameters for each block and the total of parameters.**

| Input Size | Block Name | Operation | Number of parameters |
|---|---|---|---|
| $128 \times 128 \times 1$ | Contraction 1 | Conv2d 3x3, ReLU, BatchNorm2d | 192 |
| $128 \times 128 \times 16$ | | Conv2d 3x3, ReLU, BatchNorm2d | 2,352 |
| $128 \times 128 \times 16$ | | MaxPool2d | |
| $64 \times 64 \times 16$ | Contraction 2 | Conv2d 3x3, ReLU, BatchNorm2d | 4,704 |
| $64 \times 64 \times 32$ | | Conv2d 3x3, ReLU, BatchNorm2d | 9,312 |
| $64 \times 64 \times 32$ | | MaxPool2d | |
| $32 \times 32 \times 32$ | Contraction 3 | Conv2d 3x3, ReLU, BatchNorm2d | 18,624 |
| $32 \times 32 \times 64$ | | Conv2d 3x3, ReLU, BatchNorm2d | 37,056 |
| $32 \times 32 \times 64$ | | MaxPool2d | |
| $16 \times 16 \times 64$ | Contraction 4 | Conv2d 3x3, ReLU, BatchNorm2d | 74,112 |
| $16 \times 16 \times 128$ | | Conv2d 3x3, ReLU, BatchNorm2d | 147,840 |
| $16 \times 16 \times 128$ | | MaxPool2d | |
| $8 \times 8 \times 128$ | Bottleneck | Conv2d 3x3, ReLU, BatchNorm2d | 295,680 |
| $8 \times 8 \times 256$ | | Conv2d 3x3, ReLU, BatchNorm2d | 590,592 |
| $8 \times 8 \times 256$ | | ConvTranspose2d | 262,400 |
| $16 \times 16 \times 256$ | | Concat | |
| $16 \times 16 \times 384$ | Expansion 4 | Conv2d 3x3, ReLU, BatchNorm2d | 442,752 |
| $16 \times 16 \times 128$ | | Conv2d 3x3, ReLU, BatchNorm2d | 147,840 |
| $16 \times 16 \times 128$ | | ConvTranspose2d | 65,664 |
| $32 \times 32 \times 128$ | | Concat | |
| $32 \times 32 \times 192$ | Expansion 3 | Conv2d 3x3, ReLU, BatchNorm2d | 110,784 |
| $32 \times 32 \times 64$ | | Conv2d 3x3, ReLU, BatchNorm2d | 37,056 |
| $32 \times 32 \times 64$ | | ConvTranspose2d | 16,448 |
| $64 \times 64 \times 64$ | | Concat | |
| $64 \times 64 \times 96$ | Expansion 2 | Conv2d 3x3, ReLU, BatchNorm2d | 27,744 |
| $64 \times 64 \times 32$ | | Conv2d 3x3, ReLU, BatchNorm2d | 9,312 |
| $64 \times 64 \times 32$ | | ConvTranspose2d | 4,128 |
| $128 \times 128 \times 32$ | | Concat | |
| $128 \times 128 \times 48$ | Expansion 1 | Conv2d 3x3, ReLU, BatchNorm2d | 6,960 |
| $128 \times 128 \times 16$ | | Conv2d 3x3, ReLU, BatchNorm2d | 2,352 |
| $128 \times 128 \times 16$ | Final | Conv2d 3x3, ReLU, BatchNorm2d | 1,176 |
| $128 \times 128 \times 8$ | | Conv2d 1x1, ReLU, BatchNorm2d | 55 |
| Total parameters | | | 2,315,135 |

chambers that did not match the actual anatomy of the human heart. Fig 4 displays an example of the latter, in which the segmentation mask calculated for the right ventricle (top-left in pink labeled VD) overlaps the right atrium (bottom-left in violet labeled AD). Errors like this occur because the model lacks context awareness and has only been trained on images from the systole or diastole phases of the cycle. In fact, the overlap shown in Fig 4 was detected in a frame that was quite close to the systole, implying that the model may make similar errors in this type of frame.

To address this issue, we devised a heuristic-based corrective method to automatically detect whether any of the predicted masks overlap—either above or below the actual chamber (s)—and will clip off the overlapping area of the mask if it exceeds a pre-determined threshold value. Consider the example provided in Fig 4. The method will apply a cut to the VD mask if the maximum $y$-axis value is greater than the minimum $y$-axis value of the AD mask. To

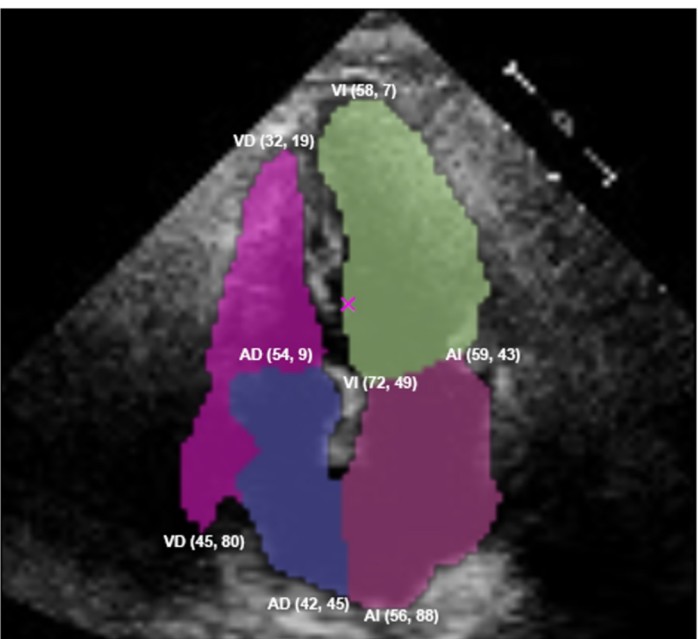

**Fig 4. An example of a mask from the right ventricle (top-left in pink labeled VD) surrounding the right atrium (bottom-left in violet labeled AD).**

determine the value, we consider the Euclidean distance between the two most extreme points of the masks and apply a cut for distances greater than 68.5 on the right masks (in this case, VD and AD) and greater than 76.5 on the left masks (the green and purple ones labeled as VI and AI, respectively). These thresholds were identified through experimenting with different values on the predicted masks prior to the method being automated in the system, which revealed that distances less than these would not benefit from being cut as the overlap is very minor, and thus should be ignored and assumed to be a regular overlap that conforms to the anatomical structure of the heart. In this example, the distance between both points is $d =$

$$\sqrt{(54-45)^2 + (9-80)^2} = 71.6$$ and will be cut as shown in Fig 5.

Furthermore, the masks can be improved for visualization purposes (for instance, to improve the contour of the VI mask on the top-right shown in green) by eroding the masks. For this, we utilised the *OpenCV* erosion morphological operation with the default parameter. Fig 6 shows the final results. This result will help clinicians to understand the location of the masks/chambers within the heart better. Furthermore, because all masks are eroded proportionally, the measurements required by clinicians (size ratios between chambers) stay consistent.

## 4. Experiments

### 4.1 U-Net

Before training our U-Net, we randomly split the database described previously using an 80/10/10 ratio. This distribution helped us follow a sequence along the training, validation and testing phases. The images from the training subset were helpful only during the training phase, deriving in good generalization. Meanwhile, the validation subset is used to calculate metrics and the loss for the validation phase, which will happen just after an epoch of the

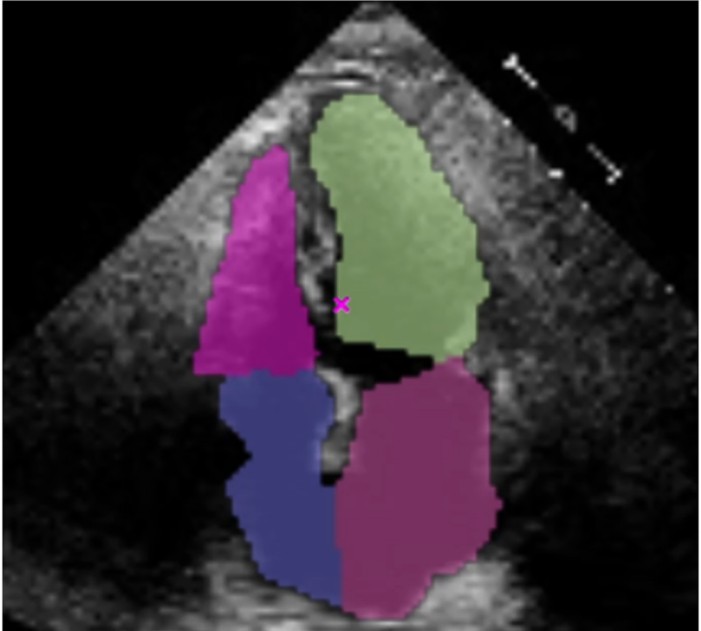

**Fig 5. Mask VD after the correction.**

training phase has finished. In other words, the validation step helped us measure the U-Net's performance. Finally, the testing subset comprised images the U-Net never visualized along the training and validation phases. The testing phase helped get a visual and numeric measure of the generalization of the CNN achieved at the end of the training.

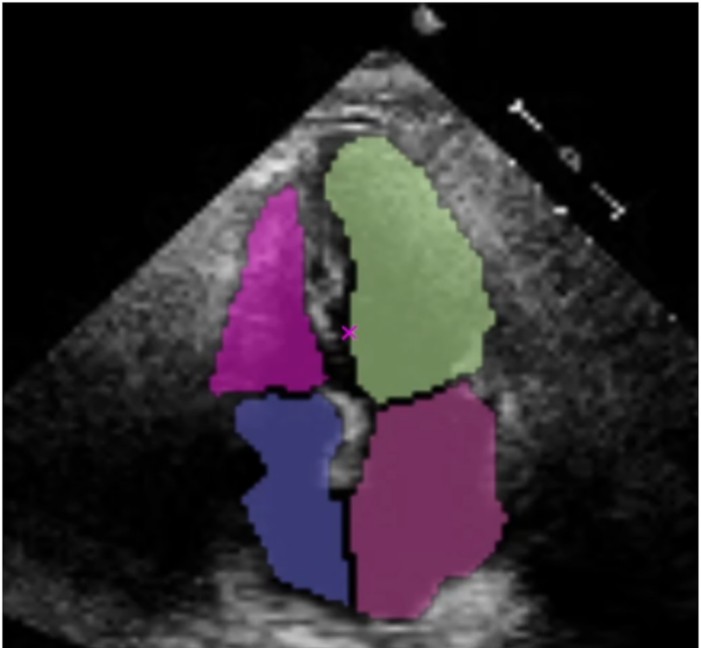

**Fig 6. The final result improved for visualization purposes.**

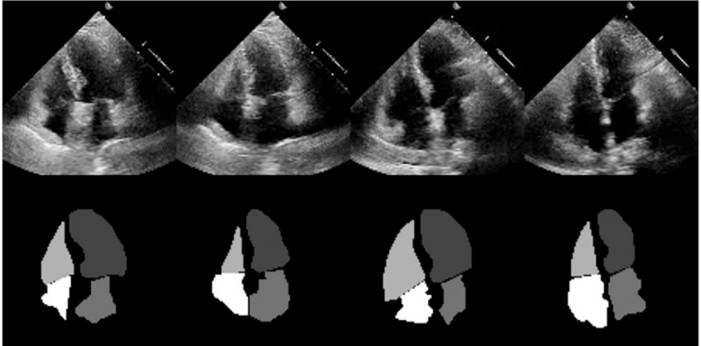

**Fig 7. Some instances from the custom dataset with their respective ground truth masks.**

To test the validity of the whole pipeline, we used a public dataset from Stanford University called EchoNet-Dynamic [39], which contains 10, 000 videos of four-chamber apical cardiac ultrasounds. Then, a proprietary dataset was created by recompiling the frames, which satisfied a set of specific requirements defined by a group of specialists from the partnering clinic. These frames were chosen with specific emphasis on the cardiac cavity visibility, image quality and sharpness. Since appropriate masks are necessary for the training, the frames were manually re-annotated with the help of the specialists. Fig 7 shows some instances from the final dataset and their respective masks.

The custom dataset was preprocessed as follows. First, the spatial resolution of each image was changed from $112 \times 112$ to $128 \times 128$; this was done to have a clear order along the subsampling path in which each image must go over. Then, a normalization from integer $(0 - 255)$ to floating values $(0 - 1)$ was applied. A training mini-batch of size eight was selected, along with data shuffling in each training epoch. Also, we set 50 training epochs with a fixed learning rate of $1e^{-3}$. The Adam optimizer was used due to its simplicity, especially since computationally efficient modules like this need few memory requirements and are suitable for large amounts of data [43]. Finally, since the application is a multiclass problem, we implemented a cross-entropy loss function.

### 4.2 Mask correction experiments

The purpose of this particular experimental validation is to demonstrate that the heuristic correction method presented in Section 3.3 is capable of improving the results obtained by the U-Net architecture (Section 3.2) in a significant manner, thus bridging the gap between human and AI judgment. To do so, an expert clinician manually labeled 218 frames of the first sequence of our dataset. Afterwards, we ran the U-Net experiments to obtain the corresponding masks. Looking at these results, a second clinician (with less experience than the first one) identified ten images where a correction of one or more masks was more needed. We then applied the heuristics correction model to all 218 images and created a new set of corrected masks.

Fig 8 shows a side-to-side comparison between the three masks from the second frame of our sequence. Notice that the correction is done in two ways; firstly, by separating the atrium from the ventricles, and secondly, by smoothing the mask edges. This, in turn, yields a better agreement between these new masks and the ground truth ones. Due to the size of the images and masks, the corrections are not noticeable at first. Therefore, Fig 9 shows a zoomed-in version that can better illustrate these differences. Notice that the left ventricle and the right

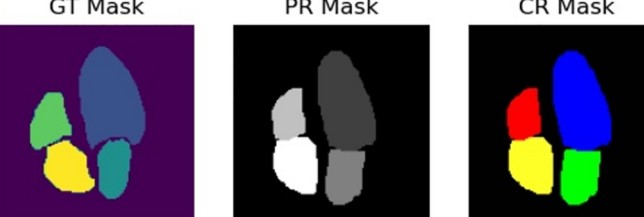

**Fig 8.** Left (GT): Mask generated by the clinician's labeling. Center (PR): Mask predicted by the model. Right (CR): Predicted mask after heuristic correction.

atrium segmentations have been smoothed (top and bottom circles respectively), while the overlap between ventricles and atriums has disappeared (left and right circles.)

To understand whether there is a gain, we calculated the Dice coefficient between all U-Net generated masks and the ground truth ones and, similarly, between all corrected masks and the ground truth ones. The first ones obtained an average Dice coefficient of 0.78, while the latter yielded an average of 0.8, implying an average 2% difference between them. More notably, the average Dice coefficient only for the masks on the frames selected by the second clinician was 0.79, while the coefficient between corrected and ground truth ones was 0.84, thus obtaining a much better gain. In the worst case (one frame that the second clinician didn't select), we had a negative difference of 25%, but we were able to notice seven frames (one of them being selected by the second clinician) where we increased over 19%

Similarly, we ran dice coefficient calculations for each of the four chamber masks. Once again, all dice coefficients were superior in all images of the sequence for the corrected masks compared to the original U-Net ones, as shown in columns 3 and 4 from Table 2. Furthermore, the differences were even better when only considering the images that were indicated to require a correction rather than the entire set (columns 5 and 6). In fact, the data of columns 5 and 6 corresponds only to images selected by a second clinician that was unsure of the prediction made by the model and then corrected with our heuristics-based algorithm. This shows that even if another clinician is unsure whether the correction should be applied or not, it is more likely that the correction yields better results rather than not applying it at all. Finally, notice that we had more success correcting masks from the right than those on the left side.

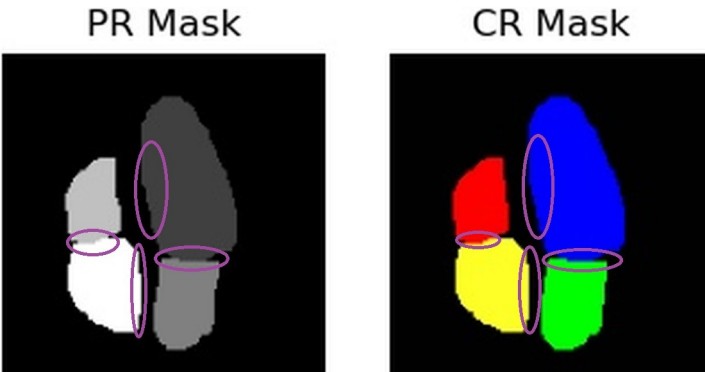

**Fig 9. Visible differences between the PR mask and the CR mask.**

**Table 2. Comparison of Dice coefficients between U-Net vs ground truth (GT) and after correction vs GT for each chamber mask.**

| Chamber | Method | Dice Coefficient average | Difference | Dice Coefficient Selected images average* | Difference |
|---|---|---|---|---|---|
| Left Ventricle | Original vs GT | 0.85 | 2% | 0.86 | 4% |
| | Corrected vs GT | 0.87 | | 0.9 | |
| Left Atrium | Original vs GT | 0.75 | 1% | 0.77 | 3% |
| | Corrected vs GT | 0.76 | | 0.80 | |
| Right Ventricle | Original vs GT | 0.73 | 1% | 0.73 | 7% |
| | Corrected VS GT | 0.74 | | 0.8 | |
| Right Atrium | Original vs GT | 0.81 | 0.4% | 0.82 | 5% |
| | Corrected vs GT | 0.814 | | 0.87 | |

* Images selected by a second clinician that was not sure if the correction should be applied or not.

## 5. Results and discussion

After 50 training epochs, the loss and metrics are obtained and shown in Figs 10 and 11 respectively. The first one compares the loss between the training phase and the validation phase, showing that both plots descend simultaneously and, with practically no significant rising peaks along the way. This is an indication of no overfitting occurring in these phases.

The second plot shows three different metrics computed in the validation phase and compared to each other, namely the Dice Coefficient (dice_coeff), the Intersection over Union (IoU) and the mean pixel accuracy. Similar to the loss, we observe a parallelism between all metrics. Notice the improved performance on the validation set is further supported by the metrics computed over the testing dataset presented in this section.

Due to our interest in knowing how the model performs over the new examples, we computed the previous three metrics over the testing dataset obtaining the following segmentation accuracy in each metric: 92% for Dice Coefficient, 85.2% for Intersection over Union and 93.5% for Mean Pixel Accuracy. On the other hand, the pixel accuracy for each cardiac cavity was computed too; with the results obtained as follows: 90.9% of pixel accuracy for the left ventricle, 90.4% for the right atrium, 86.5% for the left atrium and 86.4% for the right ventricle.

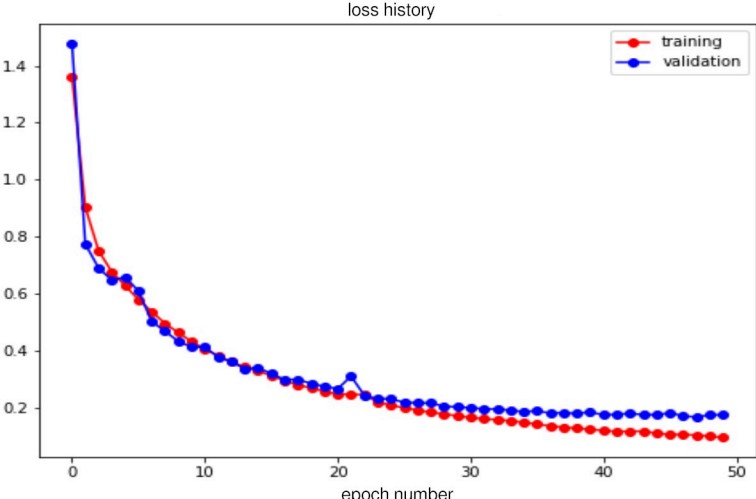

**Fig 10. Loss history for the training (red) and validation (blue) phases.**

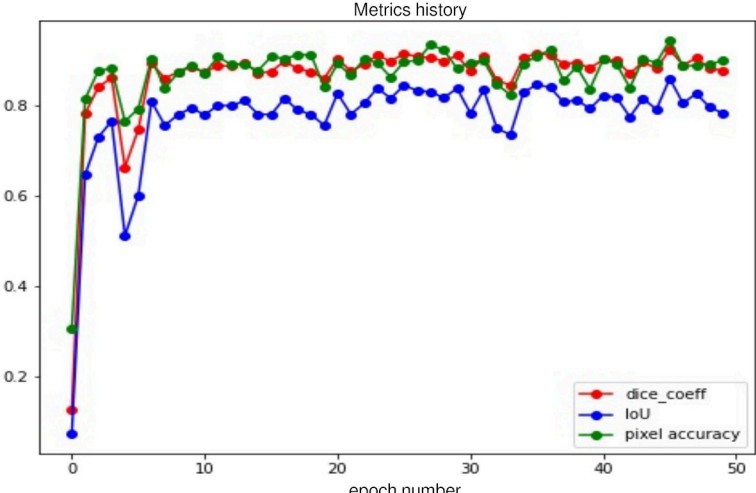

**Fig 11. Dice Coefficient (dice_coeff), Intersection over Union (IoU) and mean pixel accuracy achieved in the validation phase.**

This implies that we are capable of obtaining competitive results on images with a format different from the ones in the training set.

Another way in which we validated the performance of our proposed U-Net was through a qualitative inspection of the contours overlapped in the echocardiography images. This approach allowed us to evaluate, in a visual way, how good the segmentation was done by the deep learning algorithm. To do so, we compared the annotations of the specialists with the segmentation done by the U-Net. The testing dataset was used to generate these results, keeping in mind that all of these images were the ones that the model did not see in the training or validation phases. Four types of cases were detected, two examples and interesting characteristics by each case:

1. As a first case, we identified 15 test images with very good quality, where borders are well-defined and cavities produce a good contrast with the surfaces. Also, the cavities present regular shapes, which is typical of healthy hearts and/or good image acquisition. Fig 12

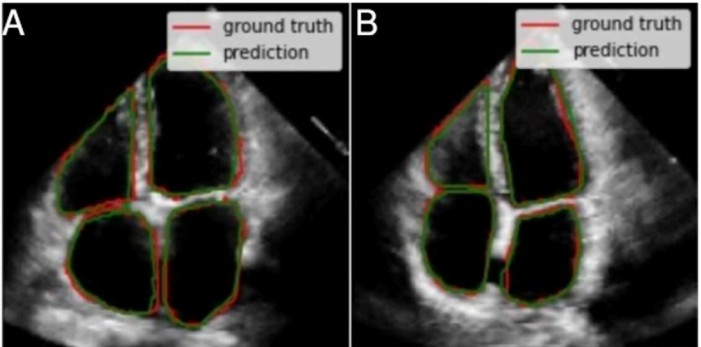

**Fig 12. Two examples from the first case where the segmentations were very close to the ground truth.** It can be proved the good quality of the images for this case (especially on the contrast between the cavities chambers and the rest of the heart structure).

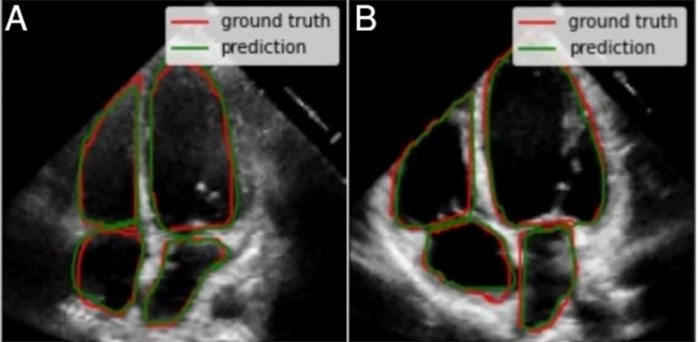

**Fig 13.** There are larger left ventricles (b) and variable shapes for the rest of the cavities and less defined borders (a) for this two examples. The model achieved good segmentations for both.

shows two examples that correspond to this first case. The segmentations are fairly close to the ground truth (i.e. annotations from the specialists).

2. For the second case, the shape of each cavity is less regular in comparison with the previous case, but images maintain proper image quality. The segmentation features do not change substantially compared to the previous case, and all of them are fairly close to the ground truth. This means that the segmentation model still achieves a high performance despite the conditions. Fig 13 also shows two examples of the 19 images that were identified in this case.

3. While in the two previous cases, the segmentations had high precision, in this third case the metrics decrease. We identified four images with worse quality, unclear borders, lower contrast and less regular cavities shapes compared to the first two cases. All of these complications are reflected in the quality of the resulting segmentations, as shown in the two examples of the Fig 14. Despite this, the segmentations can still be considered correct in a visual way by a human expert, as they follow useful patterns which approximate the cavity shapes that were learned. Furthermore, noticeable borders still remain in the images.

4. Finally, since medical annotation is a multi-observer activity, this task is subject to changes, from small and subtle to bigger and visually notable ones. This variability needs to be taken

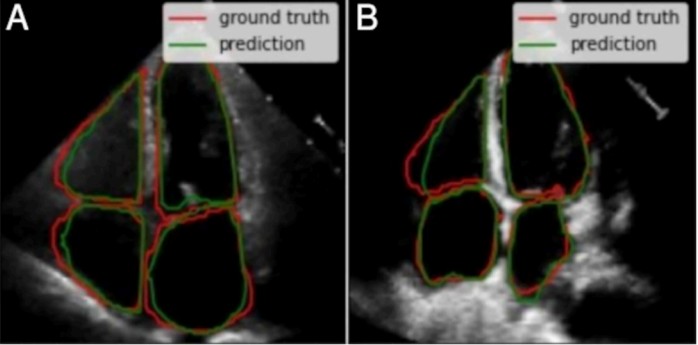

**Fig 14.** In this third case, the quality drop is notorious for this two examples: (a) and (b). This is reflected in how segmentation quality is lower compared to the two previous cases but the model does not lose shape sense and keeps them regular given the cavity.

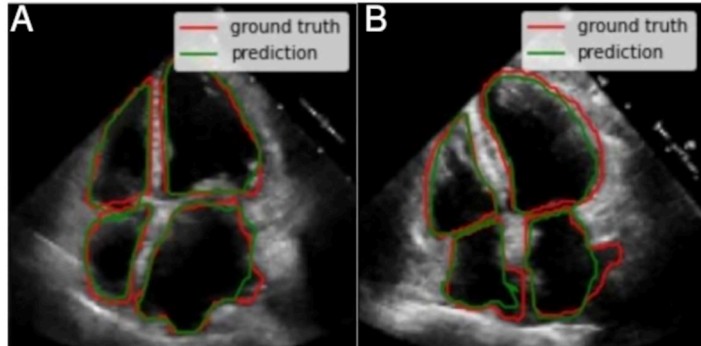

**Fig 15. Two examples of the fourth case, where annotations are incorrect and additional areas have been marked.**

into consideration when selecting the most adequate masks; this is why in this last case (that includes around five interesting and possible results) the annotations seem to be wrong and include zones which, *a priori*, shall not be considered and still, the specialist marked them. In Fig 15, we show two of these annotations and their respective segmentations made by the model.

## 6. Conclusion

In this paper, we present our latest work towards a generalized and scalable system for the analysis and segmentation of cardiac ultrasound images, which will be used to assist clinicians in diagnosing pulmonary hypertension. One of the key aspects of designing this system is the ability to cope with images with different standards, qualities, presentations, etc. Therefore, we propose a system consisting of three main stages 1) a model trained using a compendium of different ultrasound images capable of automatically segmenting the main cone of the image, thus reducing the search space and clearing out the surrounding noise 2) a robust chamber segmentation model based on the popular U-Net architecture which is capable of finding the four heart cavities and 3) a heuristics-based post-processing step which smooths the contours and corrects any overlapping. Experimental validation on a range of different images shows that our methodology has the potential to present clinicians with very accurate segmentations of the chambers, which in turn will yield more accurate measurements towards pulmonary hypertension diagnosis. Our future work is devoted to deploying this system within a perspective and prospective clinical trial and verifying the scalability of this system in low-income countries, where the images that clinics can obtain from patients have reduced quality.

## Acknowledgments

Authors acknowledge the contributions of Truong Dang, Thanh Nguyen, Rocío Aceves Millan, Beda Espinosa Caleti, Octavio Barragan García and German González Sánchez.

## Author Contributions

**Conceptualization:** Boris Escalante-Ramírez.

**Funding acquisition:** Carlos Francisco Moreno-García, Boris Escalante-Ramírez.

**Investigation:** Alan Cervantes-Guzmán, Kyle McPherson, Jimena Olveres, Carlos Francisco Moreno-García, Fabián Torres Robles, Eyad Elyan, Boris Escalante-Ramírez.

**Methodology:** Alan Cervantes-Guzmán, Kyle McPherson, Jimena Olveres, Carlos Francisco Moreno-García, Boris Escalante-Ramírez.

**Project administration:** Boris Escalante-Ramírez.

**Resources:** Boris Escalante-Ramírez.

**Software:** Alan Cervantes-Guzmán, Kyle McPherson, Carlos Francisco Moreno-García, Fabián Torres Robles.

**Supervision:** Jimena Olveres, Carlos Francisco Moreno-García, Boris Escalante-Ramírez.

**Validation:** Alan Cervantes-Guzmán, Kyle McPherson, Boris Escalante-Ramírez.

**Writing – original draft:** Alan Cervantes-Guzmán, Kyle McPherson, Jimena Olveres, Carlos Francisco Moreno-García, Boris Escalante-Ramírez.

**Writing – review & editing:** Jimena Olveres, Carlos Francisco Moreno-García, Boris Escalante-Ramírez.

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
