## [Decision Letter · Decision Letter 0]

31 Jul 2023

PONE-D-23-19397Robust cardiac segmentation corrected with heuristicsPLOS ONE

Dear Dr. Escalante-Ramírez,

Thank you for submitting your manuscript to PLOS ONE. After careful consideration, we feel that it has merit but does not fully meet PLOS ONE’s publication criteria as it currently stands. Therefore, we invite you to submit a revised version of the manuscript that addresses the points raised during the review process. Please submit your revised manuscript by Sep 14 2023 11:59PM. If you will need more time than this to complete your revisions, please reply to this message or contact the journal office at plosone@plos.org. Please include the following items when submitting your revised manuscript:A rebuttal letter that responds to each point raised by the academic editor and reviewer(s). You should upload this letter as a separate file labeled 'Response to Reviewers'.A marked-up copy of your manuscript that highlights changes made to the original version. You should upload this as a separate file labeled 'Revised Manuscript with Track Changes'.An unmarked version of your revised paper without tracked changes. You should upload this as a separate file labeled 'Manuscript'.

We look forward to receiving your revised manuscript.

Kind regards,

Anas Bilal, Ph.D.

Academic Editor

PLOS ONE

Journal Requirements:

"Newton Fund Institutional Links program project 527639907 granted by The British Council, UK, https://www.britishcouncil.org/education/he-science/newton-fund (A.C., K. M., J.O., C.F.M., F.T.R., E.E., B.E.) 

Secretaría de Educación, Ciencia, Tecnología e Innovación, CDMX, under grant 202/2019 (A.C., K. M., J.O., C.F.M., F.T.R., E.E., B.E.) https://www.sectei.cdmx.gob.mx/

Universidad Nacional Autónoma de México PAPIIT grant  IV100420 https://dgapa.unam.mx/index.php/impulso-a-la-investigacion/papiit (A.C., J.O., F.T.R., B.E.) 

Universidad Nacional Autónoma de México Programa de Becas Posdoctorales https://dgapa.unam.mx/index.php/formacion-academica/posdoc (F.T.R., B.E.) "

"Funding was provided by the Newton Fund Institutional Links program, project 527639907, granted by The British Council, UK, and by the Secretar´ıa de Educaci´on, Ciencia, Tecnolog´ıa e Innovaci´on (SECTEI), CDMX, under grant 202/2019. UNAM provided support by means of PAPIIT grants IV100420, and by the DGAPA Programa de Becas Posdoctorales. Authors would also like to acknowledge the contributions of Truong Dang, Thanh Nguyen, Roc´ıo Aceves Millan, Beda Espinosa Caleti, Octavio Barragan Garc´ıa (clinical team) and German Gonz´alez S´anchez. "

"Newton Fund Institutional Links program project 527639907 granted by The British Council, UK, https://www.britishcouncil.org/education/he-science/newton-fund (A.C., K. M., J.O., C.F.M., F.T.R., E.E., B.E.) 

Secretaría de Educación, Ciencia, Tecnología e Innovación, CDMX, under grant 202/2019 (A.C., K. M., J.O., C.F.M., F.T.R., E.E., B.E.) https://www.sectei.cdmx.gob.mx/

Universidad Nacional Autónoma de México PAPIIT grant  IV100420 https://dgapa.unam.mx/index.php/impulso-a-la-investigacion/papiit (A.C., J.O., F.T.R., B.E.) 

Universidad Nacional Autónoma de México Programa de Becas Posdoctorales https://dgapa.unam.mx/index.php/formacion-academica/posdoc (F.T.R., B.E.) "

Reviewers' comments:

Reviewer's Responses to Questions

**Comments to the Author**

1. Is the manuscript technically sound, and do the data support the conclusions?

Reviewer #1: Partly

Reviewer #2: Yes

2. Has the statistical analysis been performed appropriately and rigorously? 

Reviewer #1: No

Reviewer #2: N/A

3. Have the authors made all data underlying the findings in their manuscript fully available?

Reviewer #1: Yes

Reviewer #2: Yes

4. Is the manuscript presented in an intelligible fashion and written in standard English?

Reviewer #1: Yes

Reviewer #2: Yes

5. Review Comments to the Author

Reviewer #1: 1. Abstract is not well written.

2. If you are working on all four chambers of heart, please clearly indicate its importance, why four chambers are more significant than only considering LV (this is most important part to diagnose cardiac diseases).

3. Fig 2 is not clear, before and after segmentation, images look same.

4. Line 215, figure reference is incorrect.

5. Why is normalization required?

6. On line 228 section numbers are not mentioned.

7. In table 2, why results are shown on selected images when they are shown for all images?

8. In discussion, explain the reason of having different results in figure 12,13, 14 and 15.

9. No comparison of results with any state-of-the-art models.

10. Also mention machine learning models and image processing models for cardiac segmentation and mention papers related to them in the literature review, for example “Two-stage active contour model for robust left ventricle segmentation in cardiac MRI”. In addition to this refer more non deep learning models as well for cardiac segmentation.

Reviewer #2: In the manuscript “Robust cardiac segmentation corrected with heuristics” is presented a cardiac segmentation algorithm based on the U-NET architecture, with two additional steps to improve the quality of the results (a cone segmentation algorithm and a post-processing step.

This manuscript addresses a topic of high interest and very current. The proposed approach complements a well-known model that has already been used by other authors, with two steps that, according to the results shown by the authors, improve the accuracy of the segmentations.

In general, the paper is well written and well structured, being easy to read and understand. However, there are some points that I think can be improved, as well as some issues that need to be addressed.

My comments are as follows:

The authors claim that the proposed system implements a post-processing step that refines the shape and contours of the segmentation based on heuristics provided by physicians. However, from what I understood from the Heuristic Correction section, the correction method automatically identifies if a mask overlaps another one below/above and cuts the excess sections, based on a distance defined by the authors. In my opinion this needs a more detailed explanation.

Although the "Related Work" section is very well structured and references work in the field, in my opinion it could be improved by including more recent work. As far as I could see, only 2 works from 2023 and 1 from 2022 are referenced, the rest being earlier.

Figure 2 - To allow a faster understanding and analysis of this figure, the authors may consider identifying the images that compose it with (a), (b), ..., adding their summary description to the caption. For example (a) original image; (b) ...

Figure 3 is difficult to parse. Namely, the legends for the arrows (bottom right corner) are not legible. Authors should consider its improvement.

Table 1 should appear before “Heuristic Correction” section.

The authors define distances from which they consider that the overlapping of the masks should be removed, considering that below these values the overlaps are considered normal. Why? These assumptions need to be better substantiated.

Figures 4 and 5 are too far from the text where they are referred to and explained (with table 1 in the middle). They should be replaced.

Figure 8 - The predicted masks are very different from those generated by the clinical expert. Do the authors consider this a good/precise segmentation?

Minor comments:

Line 6 - Please, remove "." in "specialists."

Line 15 - Consider writing "... as good contrast as..." instead of "... as good a contrast as..."

Line 21 - Please, consider writing "... diagnosis results. instead of "diagnosis performance."

Line 215 - I think the authors refer to figure 7 and not figure 4.

Lines 228 and 229 - Please revise "Section ??"

Lines 77-80 - Please, revise the sentence "More recently, for 77 cardiac segmentation, a study by Dang et al. [28], [29], authors..."

Line 303 - Consider writing "... the metrics decrease. We identified,,," instead of "... the metrics decrease, we identified..."

6. PLOS authors have the option to publish the peer review history of their article (what does this mean?). If published, this will include your full peer review and any attached files.

Reviewer #1: No

Reviewer #2: No

---

## [Author Response · Author response to Decision Letter 0]

28 Sep 2023

Response to reviewers

##############

Reviewer # 1

##############

1. Abstract is not well written.

ANSWER: Abstract has been corrected for style, clarity, and structure

2. If you are working on all four chambers of heart, please clearly indicate its importance, why four chambers are more significant than only considering LV (this is most important part to diagnose cardiac diseases).

ANSWER: Dear reviewer, thank you for your concern, we added a paragraph indicating the importance of the four chambers segmentation at the end of the Introduction stating: "Furthermore, the American Society of Echocardiography and the European Association of Cardiovascular Imaging provide a set of guidelines for assessing measurements related to the four cardiac chambers. They state that these measurements are essential for evaluating cardiac function and extracting important clinical parameters".

3. Fig 2 is not clear, before and after segmentation, images look same. 

ANSWER: Thank you. The second reviewer made the same observation. We improved the Figure 2 for a more accurate and insightful interpretation

4. Line 215, figure reference is incorrect. 

ANSWER: Figure has been correctly referenced.

5. Why is normalization required? 

ANSWER: The batch normalization aids the model in achieving a smoother initialization, facilitating quicker training and even mitigating the issue of overfitting. This is explained in Section 3.2

6. On line 228 section numbers are not mentioned.

ANSWER: This has been corrected, thank you for the observation.

7. In table 2, why results are shown on selected images when they are shown for all images?

ANSWER: Table 2 shows two sets of metric calculations. In columns 3 and 4, we show the dice coefficient average and the average change (in percent) for all images on the experiment (218) in total. Meanwhile, in columns 5 and 6 we show these averages but only for the images which the clinicians marked, and thus where the adjustment was really needed. This shows that even if the correction is not required, we can still obtain an upgrade between 0.4 and 2 percent. However, if the adjustment is requested by a human observer, these results can improve to 3 to 7 percent. We have modified the table footer and included the following text to clarify this point: "Similarly, we ran dice coefficient calculations for each of the four chamber masks. Once again, all dice coefficients were superior in all images of the sequence for the corrected masks compared to the original U-Net ones, as shown in columns 3 and 4 from Table 2. Furthermore, the differences were even better when only considering the images that were indicated to require a correction rather than the entire set (columns 5 and 6). In fact, the data of columns 5 and 6 corresponds only to images selected by a second clinician that was unsure of the prediction made by the model and then corrected with our heuristics-based algorithm. This shows that even if another clinician is unsure whether the correction should be applied or not, it is more likely that the correction yields better results rather than not applying it at all. Finally, notice that we had more success correcting masks from the right than those on the left side."

8. In discussion, explain the reason of having different results in figure 12,13, 14 and 15.

ANSWER: Indeed, the images correspond to two examples for every case. We apologize for the misunderstanding generated by figures 12,13,14 and 15, which is why we decided to rephrase the descriptions of each of these figures to make our intention clearer for each of these examples.

9. No comparison of results with any state-of-the-art models. 

ANSWER: Dear reviewer, we appreciate your concern on this crucial aspect of our paper. We could not locate a state-of-the-art work that aligns with our approach. In other words, no article was found that segments four cardiac chambers in echocardiography images of adult patients. We found recent works on the segmentation of four chambers, but they focus on fetal echocardiography images, a scenario that is different than ours and is therefore not comparable. This was already pointed out in the introduction of our work (Line 29).

10. Also mention machine learning models and image processing models for cardiac segmentation and mention papers related to them in the literature review, for example “Two-stage active contour model for robust left ventricle segmentation in cardiac MRI”. In addition to this refer more non deep learning models as well for cardiac segmentation. (Alan)

ANSWER: Thank you very much for your suggestion. We have included a discussion on deformable model methods (Active Shape Models and Active Contour Models) at the beginning of Section 2.

##############

Reviewer #2 #

##############

1. The authors claim that the proposed system implements a post-processing step that refines the shape and contours of the segmentation based on heuristics provided by physicians. However, from what I understood from the Heuristic Correction section, the correction method automatically identifies if a mask overlaps another one below/above and cuts the excess sections, based on a distance defined by the authors. In my opinion this needs a more detailed explanation. 

ANSWER: Indeed, the system consists of three phases 1) Cone segmentation 2) UNET for chamber segmentation and 3) Heuristics correction of the masks generated. Indeed, in the last step the method automatically identifies if there is a mask overlap based on some adjustable parameters, and also smoothens the masks to have more realistic outputs. We have enhanced this explanation through the following text on section 3.3: "To address this issue, we devised a heuristic-based corrective method to automatically detect whether any of the predicted masks overlap - either above or below the actual chamber(s) - and will clip off the overlapping area of the mask if it exceeds a pre-determined threshold value. Consider the example provided in Figure~\\ref{fig:overlap}. The method will apply a cut to the VD mask (chamber in pink) if the maximum $y$-axis value is greater than the minimum $y$-axis value of the chamber below (i.e. the violet mask labelled AD). To determine the value, we consider the Euclidean distance between the two most extreme points of the masks and apply a cut for distances greater than $68.5$ on the right masks (in this case, VD and AD) and greater than $76.5$ on the left masks (the green and purple ones labelled as AI and VI respectively). These thresholds were identified through experimenting with different values on the predicted masks prior to the method being automated in the system, which revealed that distances less than these would not benefit from being cut as the overlap is very minor, and thus should be ignored and assumed to be a regular overlap that conforms to the anatomical structure of the heart. In this example, the distance between both points is $d=\\sqrt{{(54-45)}^2+{(9-80)}^2}=71.6$ and will be cut as shown in Figure~\\ref{fig:overlap2}." 

---------------

2. Although the "Related Work" section is very well structured and references work in the field, in my opinion it could be improved by including more recent work. As far as I could see, only 2 works from 2023 and 1 from 2022 are referenced, the rest being earlier.

ANSWER: Thank you very much for the suggestion of including more up-to-date references. We are aware of the importance of providing a study that is current, especially in rapidly evolving technological subjects. We have added as many recent references as possible (from the year 2021 onwards) that also align with the necessary relevance to our work.

-----------------

3. Figure 2 - To allow a faster understanding and analysis of this figure, the authors may consider identifying the images that compose it with (a), (b), ..., adding their summary description to the caption. For example (a) original image; (b) ...

ANSWER: We greatly appreciate your suggestion, which we gladly took and applied to Figure 2. 

4. Figure 3 is difficult to parse. Namely, the legends for the arrows (bottom right corner) are not legible. Authors should consider its improvement.

ANSWER: Thank you for this observation, we chose to upscale the Figure 3 to make it more legible.

5. Table 1 should appear before “Heuristic Correction” section.

ANSWER: Thanks, the Table 1 is now in its right place (before "Heuristic Correction" section).

6. The authors define distances from which they consider that the overlapping of the masks should be removed, considering that below these values, the overlaps are considered normal. Why? These assumptions need to be better substantiated. 

ANSWER: As mentioned in the response to the reviewer's first comment, we have enhanced this explanation in section 3.3 to address better why our assumptions are valid in the heart chamber segmentation scenario.

7. Figures 4 and 5 are too far from the text where they are referred to and explained (with table 1 in the middle). They should be replaced.

ANSWER: Dear reviewer, we would like to express our gratitude for bringing attention to this details which has been rectified and appropriately positioned near of the text where they are referred.

8. Figure 8 - The predicted masks are very different from those generated by the clinical expert. Do the authors consider this a good/precise segmentation? 

ANSWER: The original images showed a case where our segmentation was not as accurate compared to the clinician's one. We have changed the image to a better segmentation, and added the following lines to explain the new Figure 9, which is the zoom version of Fig 8: Notice that the left ventricle and the right atrium segmentations have been smoothed (top and bottom circles respectively), while the overlap between ventricles and atriums has disappeared (left and right circles).

##################

Minor comments: #

##################

Line 6 - Please, remove "." in "specialists."

ANSWER; Corrected, thanks.

------------------

Line 15 - Consider writing "... as good contrast as..." instead of "... as good a contrast as..."

ANSWER; Corrected, thanks.

------------------

Line 21 - Please, consider writing "... diagnosis results. instead of "diagnosis performance."

ANSWER; Corrected, thanks.

-----------------

Line 215 - I think the authors refer to figure 7 and not figure 4.

ANSWER: Thanks for spotting this. We have corrected the figure reference.

----------------

Lines 228 and 229 - Please revise "Section ??"

ANSWER: Corrected, and we have also added section/subsection numbers

----------------

Lines 77-80 - Please, revise the sentence "More recently, for 77 cardiac segmentation, a study by Dang et al. [28], [29], authors..."

ANSWER; Corrected, thanks.

-----------------

Line 303 - Consider writing "... the metrics decrease. We identified,,," instead of "... the metrics decrease, we identified..."

ANSWER; Corrected, thanks.

---

## [Decision Letter · Decision Letter 1]

16 Oct 2023

Robust cardiac segmentation corrected with heuristics

PONE-D-23-19397R1

Dear Dr. Ramírez,

We’re pleased to inform you that your manuscript has been judged scientifically suitable for publication and will be formally accepted for publication once it meets all outstanding technical requirements.

Kind regards,

Anas Bilal, Ph.D.

Academic Editor

PLOS ONE

Reviewers' comments:

Reviewer's Responses to Questions

**Comments to the Author**

1. If the authors have adequately addressed your comments raised in a previous round of review and you feel that this manuscript is now acceptable for publication, you may indicate that here to bypass the “Comments to the Author” section, enter your conflict of interest statement in the “Confidential to Editor” section, and submit your "Accept" recommendation.

Reviewer #1: All comments have been addressed

Reviewer #2: All comments have been addressed

2. Is the manuscript technically sound, and do the data support the conclusions?

Reviewer #1: Yes

Reviewer #2: (No Response)

3. Has the statistical analysis been performed appropriately and rigorously? 

Reviewer #1: No

Reviewer #2: (No Response)

4. Have the authors made all data underlying the findings in their manuscript fully available?

Reviewer #1: Yes

Reviewer #2: (No Response)

5. Is the manuscript presented in an intelligible fashion and written in standard English?

Reviewer #1: Yes

Reviewer #2: (No Response)

6. Review Comments to the Author

Reviewer #1: All concerns are addressed in the revised manuscript. I recommend this article for publication in this journal.

Reviewer #2: (No Response)

7. PLOS authors have the option to publish the peer review history of their article (what does this mean?). If published, this will include your full peer review and any attached files.

Reviewer #1: **Yes: **Maria tamoor

Reviewer #2: No

---

## [Editor Report · Acceptance letter]

19 Oct 2023

PONE-D-23-19397R1 

Robust Cardiac Segmentation Corrected with Heuristics 

Dear Dr. Escalante-Ramírez:

I'm pleased to inform you that your manuscript has been deemed suitable for publication in PLOS ONE. Congratulations! Your manuscript is now with our production department. 

Kind regards, 

on behalf of

Dr. Anas Bilal 

Academic Editor

PLOS ONE